# Uncovering the Nutritive Profiles of Adult Male Chinese Mitten Crab (*E. sinensis*) Harvested from the Pond and Natural Water Area of Qin Lake Based on Metabolomics

**DOI:** 10.3390/foods12112178

**Published:** 2023-05-29

**Authors:** Yuhui Ye, Yulong Wang, Pengyan Liu, Jian Chen, Cunzheng Zhang

**Affiliations:** Jiangsu Key Laboratory for Food Quality and Safety-State Key Laboratory Cultivation Base of Ministry of Science and Technology, Institute of Food Safety and Nutrition, Jiangsu Academy of Agricultural Sciences, Nanjing 210014, China; yyh@jaas.ac.cn (Y.Y.); yllzzn@sina.com (Y.W.); 20210047@jaas.ac.cn (P.L.); chenjian@jaas.ac.cn (J.C.)

**Keywords:** harvest time, metabolomic profiling, polyunsaturated fatty acid, pond culture

## Abstract

*E. sinensis*, normally harvested in October and November, is an economic aquatic product in China. Pond culture has been widely applied for the production of *E. sinensis*, wherein a stable food supply for crabs is provided. In order to improve the nutritional quality of *E. sinensis* products, this study evaluated the effect of the local pond culture on the nutritive profiles of *E. sinensis* and screened out the best harvest time for the nutrient-rich crabs, thereby guiding the local crab industry to improve its aquaculture mode and harvest strategy. The results indicated that pond culture enhanced the levels of protein, amino acids, and specific organic acid derivatives, and reduced the levels of peptides and polyunsaturated fatty acids (PUFAs). Compared with *E. sinensis* harvested in October, peptide levels were significantly increased, whereas sugar, phenolic acid, and nucleotide levels were decreased in those harvested in November. Overall, the study revealed that the nutritive profile of the pond-reared *E. sinensis* was significantly modulated by a high-protein diet, thus lacking the diversity of metabolites. Additionally, October could be more appropriate for harvesting *E. sinensis* than November.

## 1. Introduction

Chinese mitten crab (*E. sinensis*) is an important aquatic product native to the East Pacific coast of China and the Korean Peninsula [1]. Although *E. sinensis* has increasingly entered into Europe and America, China has accounted for the most of its production in the world, reaching 770,000 tons in 2020 [2,3,4]. Jiangsu province of China has the largest crab-producing water area, belonging to the Yangtze River, annually producing half of *E. sinensis* over the country [3].

Qin Lake is located in the north of Jiangsu province and flows as a tributary of Yangtze River. With a subtropical monsoon climate, the average temperature of Qin Lake is 15.3 °C, and the mean annual sunshine duration is 2075 h. Moreover, the lake water has a high dissolved oxygen content (5.5–8 mg/L) and relatively constant pH (7.2–8.6), providing an ideal environment to the local aquaculture [5]. The local people from Qin Lake harvest *E. sinensis* using a unique tool, ‘Duan’, which is a set of circle-shaped bamboo weirs located in the lake. Small and immature crabs normally pass around the weir circle; however, strong crabs are able to climb over the weir, thus falling down to the center of the weir circle and being captured afterwards [6]. Owing to this special harvest strategy and the ideal lake environment, *E. sinensis* harvested from Qin Lake may possess high nutritional quality. 

Protein and fat are two of the major fundamental nutrients in crabs. Crab protein is considered nutritious, owing to the high level of essential amino acids, which not only contribute to the flavor but also exerted various biological effects after being absorbed and metabolized by the human gastrointestinal tract [7]. Fat is nutritionally significant in crustaceans, where SFA (saturated fatty acid), MUFA (monounsaturated fatty acid), and PUFA (polyunsaturated fatty acids) are the key biomarkers for human health. SFA was proven to elevate plasma cholesterol and could be involved in inflammation response [8,9]. However, MUFA and PUFA are able to reduce the level of low-density lipoprotein (LDL) and exhibit beneficial effects on the risk markers of cardiovascular disease, including inflammation, blood pressure, and vascular function [10,11]. 

Besides protein and fat, other delicate but important nutritional compounds have been uncovered using a metabolomics approach, which was an analytical profiling technique for measuring and comparing large numbers of metabolites present in cells, organs, tissues, or biofluids. As a high-throughput and high-sensitivity technique, metabolomics has been widely adopted recently for the characterization of aquatic products’ nutritional quality. For example, the flesh quality of grass carp was evaluated by a UPLC-QTOF/MS-based metabolomics study [12]. Integrated metabolomic and gene expression analyses were conducted for analyzing the meat quality of large yellow croaker [13]. Therefore, metabolomics is an effective method for evaluating aquatic products’ nutritional quality. Although the metabolomics data of *E. sinensis* has been collected recently [14], there has been no systematic metabolomics study comparing the nutritive profiles of the pond-reared and naturally produced *E. sinensis*. Normally, crabs reared in a pond are offered a stable food supply, but at the same time, they have limited space for living and reproducing. In contrast, naturally produced crabs are nourished by the lake’s eco-system, whereas they should face the environmental changes independently. Hence, the way in which these different aquaculture modes affect the nutritive profiles of *E. sinensis* remains to be elaborated. Further, October and November have been regarded as the best season for crab harvesting in China [3], but no report is available that has investigated the delicate difference of the nutritional quality of crabs harvested between these two months. 

Therefore, the present study aimed to decipher the nutritive profiles of *E. sinensis* harvested from the pond and natural water area of Qin Lake, as well as to compare the different nutritional compositions between *E. sinensis* harvested in October and November, thereby providing comprehensive metabolomics information for the aquaculture of *E. sinensis* and thus facilitating the production of *E. sinensis* with superior nutritional quality.

## 2. Materials and Methods

### 2.1. Sample Preparation and Sampling

The crabs were reared in a 0.8 ha water pond and fed a high-protein diet (91% of dry matter, 56% of crude protein, 9% of crude lipid, and 13% of ash) twice per day. However, the crabs from the natural water area of Qin Lake, China (32°31′11.47″–32°42′56.74″ N, 119°53′16.94″–120°16′14.52″ E), were mostly dependent on natural food, such as algae, zoobenthos, and planktons. Twenty-one male crabs (200.0 ± 10.0 g) were collected from the pond in October 2021 (P-M10) and November 2021 (P-M11), respectively. Twenty-one male crabs (200.0 ± 10.0 g) were collected from natural water area in October 2021 (N-M10) and November 2021 (N-M11), respectively. Then, the crabs were stored in a plastic foam box with a small amount of water and thereafter were transported to the laboratory within an hour. After that, the crabs were cleaned with water and sacrificed, and the crab meat, including breast, feet and claw meat, and crab cream, including the hepatopancreas and gonad, were picked by hand. All the samples were frozen at −20 °C before analyses. The 21 crabs of each group were divided into two parts: 3 crabs were chosen for the proximate composition, free amino acid, and fatty acid composition analyses, and the rest of the 18 crabs were analyzed by the metabolomics approach. The edible part of the crab, including the meat and cream, was mixed before analysis. Therefore, 18 edible parts of crabs were obtained, and they were further divided into 6 biological replicates by randomly mixing every 3 edible parts together. 

### 2.2. Proximate Composition Analysis

The moisture content and protein content were analyzed on the basis of Association of Analytical Chemists Procedures [15]. For the ash content measurement, 0.5 g of the sample was added to the crucible and combusted at 550 ± 25 °C for 30 min. The ash was then cooled down and weighed. The ash content (%) was finally presented as the ratio of the ash weight to the sample weight. The total lipid was extracted with absolute ether by a Soxhlet apparatus (BSXT-02, Yuming Co., Ltd., Shanghai, China). A total of 1 g of the sample was added to the evaporating dish and removed the moisture via evaporation, and then we transferred the dried sample into a filter paper pocket. The paper pocket was then transferred to the Soxhlet apparatus, and absolute ether was added to extract the total lipid. The whole apparatus was run under a water bath for 6–8 h. After extraction, the total lipid was weighed, and the lipid content (%) was calculated by dividing by the original sample weight.

### 2.3. Amino Acid Determination

A total of 5 mL of 10% sulfosalicylic acid (304851-84-1, Sigma, Darmstadt, Germany) was added to the crushed crab meat or cream samples. Then, the solution was centrifuged at 12,000× *g* for 20 min. We collected the supernatant and added 1 mL of petroleum ether (101316-46-5, Sigma, Darmstadt, Germany). After 1 h, we removed the petroleum ether and obtained the solution in the bottom. Lastly, the solution was filtrated by a 0.22 μm syringe filter and loaded to an automatic amino acid analyzer (LA 8080, Hitachi, Japan) equipped with a Hitachi high-performance cation exchange column. 

### 2.4. Fatty Acid Composition

A total of 20 mL of chloroform/methanol solution (*v*/*v*, 2:1, Sigma, Darmstadt, Germany) was added to the 1.0 g sample. After 2 h, we removed the chloroform by a rotatory evaporator (RE-52AA, Yarong Co., Ltd., Shanghai, China). Then, 3 mL of 0.5 mol/L sodium hydroxide methanol solution was added, and the mixture was heated to 50 °C for 20 min by a thermostatic water bath (HH-1, Yuexin Instrument Manufacturing, Co., Ltd., Changzhou, China). Next, methanolic boron trifluoride (3 mL, 13%, Sigma, Darmstadt, Germany) was added, and the mixture was heated again at 85 °C for 30 min. Finally, after cooling down to the room temperature, 1 mL of n-hexane was added, and the mixture was filtered through a 0.45 μm syringe filter. The solution was applied to an Agilent 6890 gas chromatograph equipped with an HP-5.5% phenyl methyl Si-loam capillary column (30.0 m × 25 mm, Agilent 19091J-413, Santa Clara, CA, USA).

### 2.5. Wide Targeted Metabolomics Analysis 

A total of 20 mg of each replicate (6 biological replicates in each experimental group) was weighed and homogenized (30 HZ) for 20 s with a steel ball. The mixture was centrifuged (3000 rpm, 4 °C) for 30 s. Then, 400 μL of 70 % methanol was added, and the solution was shaken (1500 rpm) for 5 min and placed on ice for 15 min. After that, the solution was centrifuged at 12,000 rpm, 4 °C for 10 min, and 300 μL of the supernatant was obtained. Finally, the supernatant was centrifuged again (12,000 rpm, 4 °C) for 3 min, and the newly obtained supernatant was collected for LC-MS/MS analysis, where a LC-ESI-MS/MS system (UPLC, ExionLC AD; MS, QTRAP^®^ System) was applied. The analytical conditions were as follows: UPLC column, Waters ACQUITY UPLC HSS T3 C18 (1.8 μm, 2.1 mm × 100 mm); column temperature, 40 °C; flow rate, 0.4 mL/min; injection volume, 5 μL; solvent system, water (0.1% formic acid): acetonitrile (0.1% formic acid); gradient program, 95:5 *v*/*v* at 0 min, 10:90 *v*/*v* at 10.0 min, 10:90 *v*/*v* at 11.0 min, 95:5 *v*/*v* at 11.1 min, 95:5 *v*/*v* at 14.0 min. Linear ion trap (LIT) and triple quadrupole (QQQ) scans were acquired on a triple quadrupole-linear ion trap mass spectrometer (QTRAP), QTRAP^®^ LC-MS/MS System, equipped with an ESI Turbo Ion-Spray interface, operating in positive and negative ion modes and controlled by Analyst 1.6.3 software (Sciex). The ESI source operation parameters were as follows: source temperature, 500 °C; ion spray voltage (IS), 5500 V (positive) or −4500 V (negative); ion source gas I, gas II, and curtain gas were set at 50, 50, and 25 Psi, respectively; the collision gas was set as high. Instrument tuning and mass calibration were performed with 10 and 100 μmol/L polypropylene glycol solutions in QQQ and LIT modes, respectively. 

### 2.6. Statistical Analysis

The significant differences among samples were analyzed by one-way ANOVA with SPSS statistics package software (version 25.0, Chicago, IL, USA), using Tukey’s range test under a significant level of *p* < 0.05. In terms of MS data analysis, unsupervised PCA (principal component analysis) was performed by statistics function prcomp within R (www.r-project.org, accessed on July 2022). The data were unit variance scaled before unsupervised PCA. Variable importance of the projected (VIP) values and the permutation plots were extracted from orthogonal partial least squares discriminant analysis (OPLS-DA) result, which was generated using the R package MetaboAnalystR. The data was log transformed (log2) and mean centered before OPLS-DA. Differential metabolites were selected on the basis of VIP > 1, absolute Log_2_FC (fold change: the ion intensity of the metabolite in experimental group/the ion intensity of the metabolite in control group) > 2, and *p* < 0.05.

## 3. Results

### 3.1. Proximate Composition of the E. sinensis

The proximate composition of the edible part of *E. sinensis* (meat and cream) is presented (Appendix A). P-M11 showed the highest protein content in meat samples, followed by P-M10, N-M11, and N-M10. N-M11, P-M11, and P-M10 had significantly higher fat content compared to N-M10 (*p* < 0.05). For crab cream, the pond-reared crabs (P-M10 and P-M11) also had the most proteins, and crabs harvested in November possessed higher protein content compared to those harvested in October. The lowest fat content was observed in N-M11; however, no significant difference was found among the other groups (*p* < 0.05).

### 3.2. Amino Acid Determination 

The pond-reared crabs had a significantly higher amount total amino acid in meat compared to crabs harvested in a natural water area (*p* < 0.05, Table 1). 

Glu, Gly, Ala, Val, Phe, Lys, His, and Arg were the most significant amino acids accumulated in the meat of pond-reared crabs. The crabs harvested in November possessed a greater level of total amino acids in cream compared to those harvested in October (*p* < 0.05), which was mostly due to the increased levels of Glu, Gly, and Ala.

### 3.3. Fatty Acid Composition

For the crab meat, the pond-reared crabs had significantly higher MUFAs and lower SFAs and PUFAs than crabs from a natural water area (*p* < 0.05, Table 2).

Crabs harvested in October had significantly more SFAs and less MUFAs and PUFAs than those harvested in November (*p* < 0.05). For the cream samples, the levels of PUFAs were less in the crabs harvested in November when compared to those harvested in October (*p* < 0.05). Further, the pond-reared crabs had significantly more SFAs and less PUFAs compared to those from a natural water area (*p* < 0.05).

### 3.4. Overview of the Metabolic Profile of E. sinensis

A widely targeted metabolomic analysis was performed on the meat and cream (mixture) of *E. sinensis*. In total, 1920 metabolites were detected, including 680 amino acids and their metabolites; 158 organic acids and their metabolites; 156 heterocyclic compounds; 151 fatty acyls; 129 benzene and substituted derivatives; 117 nucleotides and their metabolites; 95 glycerophospholipids; 91 aldehyde, ketones, and esters; 84 alcohol and amines; 60 carboxylic acids and derivatives; 34 hormones and hormone-related compounds; 31 glycerolipids; and 46 undefined organic compounds and other minor components (less than 1% of total metabolites) (Figure 1A).

Unsupervised PCA was conducted to compare the differences of the metabolite profiles among all the groups, where six biological replicates were clustered together by different colors (Figure 1B). The first principal component (PC1) was with a 29.41% variance contribution value, and P-M10 and P-M11 were separated by line X = 0 as distinct groups. However, N-M10 and N-M11 were relatively similar, sharing large areas of overlap. On the basis of the second principal component (PC2), the line Y = 0 separated groups between crabs from the natural water area (N-M10, N-M11, Y > 0) and the pond (P-M10, P-M11, Y < 0). The result indicated that the pond culture significantly altered the metabolites of *E. sinensis*. In addition, the harvest time (maturity of crabs) remarkably affected the metabolites in pond-reared crabs, whereas metabolites in crabs from a natural water area were relatively similar between October and November.

### 3.5. Differential Metabolites Analysis

#### 3.5.1. Effect of Pond Culture on the Metabolites of *E. sinensis*

The differential metabolites were analyzed according to pairwise comparisons, using the OPLS-DA model. The permutation tests of OPLS-DA models are shown in the Appendix A), which reflects the robustness of the model. A further differential metabolites screening was applied, wherein metabolites with a fold change > 2 (upregulation) or <0.5 (downregulation), VIP > 1, and *p* < 0.05 were regarded as significant differential metabolites. In the present study, there were 275 upregulated and 174 downregulated metabolites in P-M10 compared with N-M10 (Figure 2A,C).

Six organic acid derivatives were among the top 10 upregulated metabolites, including stachydrine, methylsuccinic acid, and glutaric acid, whereas the top 10 downregulated metabolites mostly contain small peptides. In November, there were more downregulated metabolites (373) than upregulated metabolites (66) observed in P-M11 compared to N-M11 (Figure 2B,D). Similar to the results from October, peptides were the most downregulated metabolites (127 out of 373), followed by glycerophospholipids and heterocyclic compounds. Organic acid derivatives were also the most upregulated metabolites. This showed that the pond culture exhibited a similar effect on the nutritive profiles between *E. sinensis* harvested in October and November. In addition, the Venn diagram illustrates the common metabolites shared between P-M10 and P-M11, revealing the key metabolites modulated by pond breeding (Figure 2E).

#### 3.5.2. Effect of Harvest Time (Maturity) on the Metabolites of *E. sinensis*

There were 165 differential metabolites identified in N-M11 compared to N-M10 (Figure 3A,C).

The most upregulated metabolites included 5,6-epoxy-eicosatrienoic acid (EET), N-stearamide, peptide HQTE, hypericin, and palmitoylethanolamide (PEA). Urocanic acid, trigonelline, and urea, followed by several sugars, including D-trehalose, lactose, lactulose, and maltose, were the most downregulated metabolites (Figure 4).

Regarding on the pond-reared crabs, the metabolites were significantly different between October and November, due to approximately one-third of the metabolites (674 out of 1908) being identified as differential metabolites according to our screening threshold (Figure 3B,D). Small peptides and carnitine (C18 and C14) were the most upregulated metabolites, whereas phenolic acids (8-paradol and coniferaldehyde) and nucleotides (2′-deoxyuridine and deoxyguanosine) were significantly downregulated (Figure 4). Moreover, 57 key metabolites affected by the maturity of crabs were identified (Figure 3E).

## 4. Discussion

The production of *E. sinensis* in China experienced a rapid increase, from 17,500 tons in 1993 to 77,000 tons in 2020, which brought in huge economic benefit, reaching a gross annual output value of more than 10 billion dollars [16]. Although a huge business success occurred for this large-scale crab industry, the nutritional quality of the products, which may be compromised for the yield, possessed great potential to be improved. The present study firstly focused on the aquaculture mode and therefore intended to figure out how the traditional pond culture at Qin Lake affected the nutritive profiles of *E. sinensis*. Secondly, the study elucidated the difference of nutritive profiles between crabs harvested in October and November, thereby screening out the best harvest time, in order to facilitate the production of *E. sinensis* with superior nutritional quality. As a metabolomics-based study, it should be noted that the crabs were transported to the laboratory freshly after harvesting without any freezing or drying steps. Therefore, the metabolites kept their best activities during the following extraction step.

The characterization of samples, regarding the proximate composition of the pond-reared and naturally produced crabs, was firstly determined. The pond culture showed a significant effect in terms of increasing the protein level of both crab meat and cream. Further, P-M10 and P-M11 contained significantly higher fat content than N-M11 in the crab cream (*p* < 0.05), which was in agreement with He et al. [17], where pond culture exhibited a significant growth-promoting effect on crabs, including enhancement of both fat and protein content. Pond culture was often applied as a traditional aquaculture mode, in order to ensure a stable growth of the aquatic products, thus facilitating a high yield and thereby creating economic values. However, people nowadays focused more on their health and began to consider whether the consumption of this unnaturally cultured aquatic products would lead to abnormal nutritional uptake or even health problems. Contrary to natural lake water, the water in the pond flowed less, and therefore may have contained less oxygen and fewer small organisms, although enough food for the crabs was supplemented. Several recent studies indicated that the strong crabs reared in ponds were not always associated with superior nutrition, suggesting that various kinds of metabolites, rather than protein and fat, also contributed to the crabs’ nutritive profile and exhibition of high nutritional value [18,19,20]. This raised the need for a wide-targeted metabolomics study to decipher the subtle difference of the nutritive profiles between pond-reared and naturally produced crabs. Regarding the free amino acid, pond-reared crabs accumulated more Glu, Gly, Ala, Val, Phe, Lys, His, and Arg than naturally produced crabs. Additionally, the sum levels of essential amino acids in pond-reared crabs were significantly higher than those in naturally produced crabs (*p* < 0.05). A similar result was observed by He et al. [21], who found that almost all the amino acid levels were elevated in the pond-reared swimming crabs (Portunus trituberculatus). The amino acids in crabs not only provided flavor but also participated in amino acid metabolism after being absorbed by the human body, exerting numerous biological processes. For example, Glu/Gln and Asp/Asn were involved in nitrogen assimilation, synthesis of other amino acids and nitrogenous compounds, and carbon/nitrogen partitioning [22]. Lys was the major building blocks for growth factors and played important roles in cell division and wound-healing processes. Deficiency in Lys could cause impaired growth performance, immune response, and inflammation in grown-up grass carp [23]. Further, Lys and Met were the precursors for the synthesis of carnitine, which was able to convert fat into energy and suppress oxidative stress [24]. Additionally, regarded as a dietary essential amino acid and the precursor for various hormones, His showed great importance in kidney function, gastric secretion, the immune system, and neurotransmission [25]. Overall, the pond culture enriched these important amino acids in *E. sinensis*, owing to its stable supply of protein-rich food.

Fatty acids also played important roles in the energy metabolism and regulation of blood glucose [26]. However, in contrast to free amino acids, the pond-reared crabs had significantly less PUFAs than crabs from a natural water area, both in terms of the crab meat and cream (*p* < 0.05). Similar results were observed by Kong, Cai, Ye, Chen, Wu, Li, Chen, and Song [18], who indicated that the snails in the natural water area served as an extra source of PUFA for crabs, leading to an increase in PUFAs in the naturally produced crabs. Moreover, the naturally produced crabs had a larger space for living and traveling, thus being likely to develop more muscle, wherein the stearoyl-CoA desaturase 1 (SCD1) from crab muscle was able to catalyze SFA into PUFA [27]. PUFAs, including omega-3 and omega-6 fatty acids, could maintain cell membrane fluidity and improve vascular endothelial cell function, as well as being important to the regulation of obesity, metabolic disease, cardiovascular and immune functions, and neuropsychiatric disorders [28]. However, pond culture led to decreased PUFAs in the present study, which may have been due to the deficiency of snails in the pond and the reduced muscle content in crabs.

The organic acid derivatives stachydrine, methylsuccinic acid, and glutaric acid were significantly upregulated in the pond-reared crabs compared to those cultured in a natural water area. Interestingly, stachydrine was a potent bioactive compound, having demonstrated various activities against tissue fibrosis, cardiovascular diseases, cancers, uterine disease, and inflammation [29]. However, methylsuccinic acid is a kind of urinary metabolite that has been regarded as one of the potential biomarkers for nephrotoxicity [30]. Glutaric acid could lead to apoptosis in immature oligodendrocytes and has been associated with glutaryl-CoA dehydrogenase deficiency, which is an inherited metabolic disease [31]. In addition, the levels of peptides were significantly downregulated in the pond-reared crabs, which could have been due to the amino acid metabolism, where peptides were consumed and free amino acids were produced.

Therefore, the sufficient food provided by ponds resulted in the accumulation of protein, free amino acids, and specific organic acids derivatives. However, it also led to the decreased release of peptides and PUFAs compared to naturally produced crabs. These results revealed that the pond culture significantly affected the nutritive profile of *E. sinensis* via the modulation of mostly protein, peptide, and amino acid levels. This could have been due to the food supplied by pond being mostly protein-based; however, naturally produced *E. sinensis* had access to various organisms in the lake, e.g., snails, zoobenthos, and planktons, leading to a more diverse nutritive profile, including elevated levels of PUFAs. Although snails, small fish, and animal-based freeze-dried powder could also be added to the pond, it not only raised the cost for crab rearing but also brought contaminants to the water, and therefore the local pond owners only used the cost-effective plant-based protein to rear *E. sinensis*. This implied that the pond culture was still relatively primitive at the Qin Lake area, although the environment there was highly suitable for crab rearing. The introduction of animal-based feed, combined with the improvement of a pond water circulation (inlet and drainage) system, growing aquatic plants with self-purification capacity, or reducing the culture density, could be ways to enhance the nutritional quality of the pond-reared crabs [16].

The nutritional quality of crabs was not only decided by the aquaculture mode but also affected by the harvest time when crabs with different maturities were collected. The local farmers at Qin Lake typically harvest crabs through the entire crab season, i.e., October, November, and possibly the beginning of December, thereby obtaining as many crabs as possible for selling. However, it was extremely important for the large-scale crab industry to focus on the best representative products instead of the yield, and therefore a comparison of the crab nutritive profiles between October and November was conducted.

In the present study, 5,6-EET significantly accumulated in the naturally produced crabs harvested in November, when compared to those harvested in October (*p* < 0.05). EETs were cytochrome-P450-epoxygenase-derived metabolites that play important biological roles in humans. Reports showed that they could not only improve the vasodilatation but also modulate insulin sensitivity, therefore being regarded as a potential therapeutic target for diabetes and ischemic cardiomyopathy [32,33]. Two amines, N-stearamide and PEA, were also remarkably upregulated in N-M11. However, they showed distinct bioactivities. N-stearamide was a potential biomarker for disordered fatty acid metabolism related to hepatic cirrhosis [34], whereas PEA was a ligand of peroxisome proliferator-activated receptors (PPAR-α) that exerted neuroprotective, anti-inflammatory, analgesic, and lipid-regulating effects [35,36]. Surprisingly, hypericin, a plant-originated bioactive compound that possesses anti-depressive, anti-neoplastic, anti-tumor, and anti-viral activities [37], was enriched in N-M11. This might provide evidence that the naturally produced crab consumes plant for survival in November. However, sugars, including D-trehalose, lactose, lactulose, and maltose, were significantly downregulated in N-M11. The consumption of carbohydrates provides crabs with enough energy to live through the cold weather in November.

Different to the naturally produced crabs, the pond-reared crabs had a stable energy supply during the entirety of November. This led to the significant increase in peptide levels in P-M11 compared to P-M10, as the protein-based food was largely digested. However, significant amounts of phenolic acids and nucleotides, which were not only important nutrients but also flavor compounds, were downregulated in P-M11. This was in line with studies that indicated that the total free amino acids, hydroxyeicosatrienoic acid, umami 5′-nucleotide compounds, and PUFAs were decreased in pond-reared crabs [18,20]. It was speculated that the strategy for a pond-reared crab to survive in November could be in consuming sufficient food and staying still in the pond, leading to a significant increase in peptide levels for energy provision. Overall, it could be concluded that October was more appropriate for harvesting *E. sinensis* than November, due to the elevated levels of sugars, phenolic acids, and nucleotides. Therefore, the local farmers could consider reducing the duration of the harvest time, instead of harvesting throughout the whole crab season, while finding a balance between the nutritional quality and the yield of crabs. In addition, controlling the fluctuation of water temperature was found to be crucial for the nutritional quality of *E. sinensis* in cold November.

## 5. Conclusions

To conclude, pond culture enhanced the levels of protein, free amino acids (Glu, Gly, Ala, Val, Phe, Lys, His, and Arg), and organic acid derivatives (stachydrine, methylsuccinic acid, and glutaric acid) and reduced the levels of peptides and PUFAs in *E. sinensis*. For pond-reared *E. sinensis* harvested in November, the levels of peptides were elevated, whereas phenolic acid (8-paradol and coniferaldehyde) and nucleotide (2′-deoxyuridine and deoxyguanosine) levels were declined, compared with those harvested in October. Above all, the metabolic profile of the pond-reared *E. sinensis* was significantly modulated by the high-protein diet, thus lacking the diversity of metabolites, especially PUFA levels. October could be more appropriate for harvesting *E. sinensis* in terms of crabs’ nutritional quality. Therefore, our study facilitated the development of a high-end *E. sinensis* product by providing new understandings on the pond culture and the harvest strategy of crabs.

## Figures and Tables

**Figure 1 foods-12-02178-f001:**
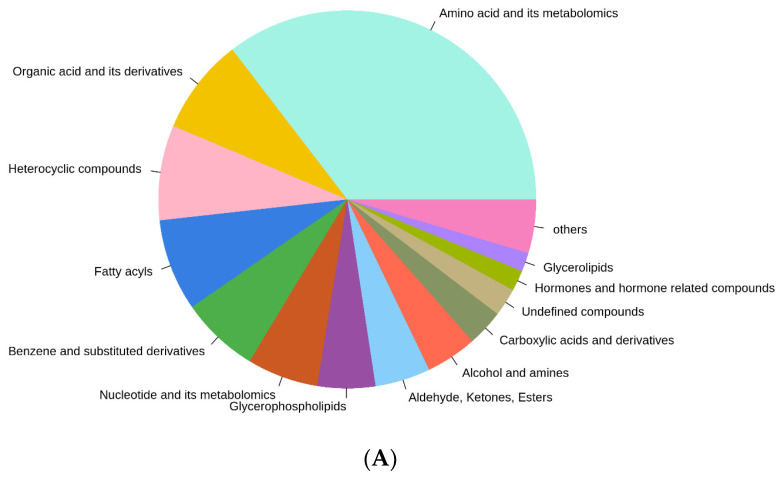
The overview of the metabolomic profile (**A**) and principal component analysis of *E. sinensis* harvested from the pond and natural water area of Qin Lake (**B**). Amino acid and its metabolomics: 35.42%; organic acid and its derivatives: 8.23%; heterocyclic compounds: 8.12%; fatty acyls: 7.86%; benzene and substituted derivatives: 6.72%; nucleotide and its metabolomics: 6.09%; glycerophospholipids: 4.95%; aldehyde, ketones, and esters: 4.74%; alcohol and amines: 4.38%; carboxylic acids and derivatives: 3.12%.

**Figure 2 foods-12-02178-f002:**
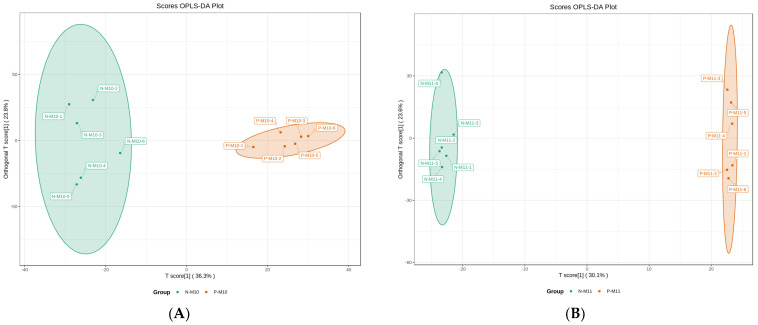
The score plots of OPLS-DA pairwise comparisons between N-M10 and P-M10 (**A**), N-M11 and P-M11 (**B**). Volcano plots of the differential metabolites between N-M10 and P-M10 (**C**), N-M11 and P-M11 (**D**). Venn diagram of differential metabolites (**E**). The T score of the score plot indicates the degree of variations between groups, and the orthogonal T score represents the degree of variations within the group. Percentages indicate how well a component explains the dataset. The volcano plot illustrates the distribution of the up-/downregulated differential metabolites. In the Venn diagram, each comparison group is illustrated as a circle, and the number in it indicates the number of differential metabolites. The number in the overlapped area of the circles indicates the number of the common differential metabolites among all the comparison groups.

**Figure 3 foods-12-02178-f003:**
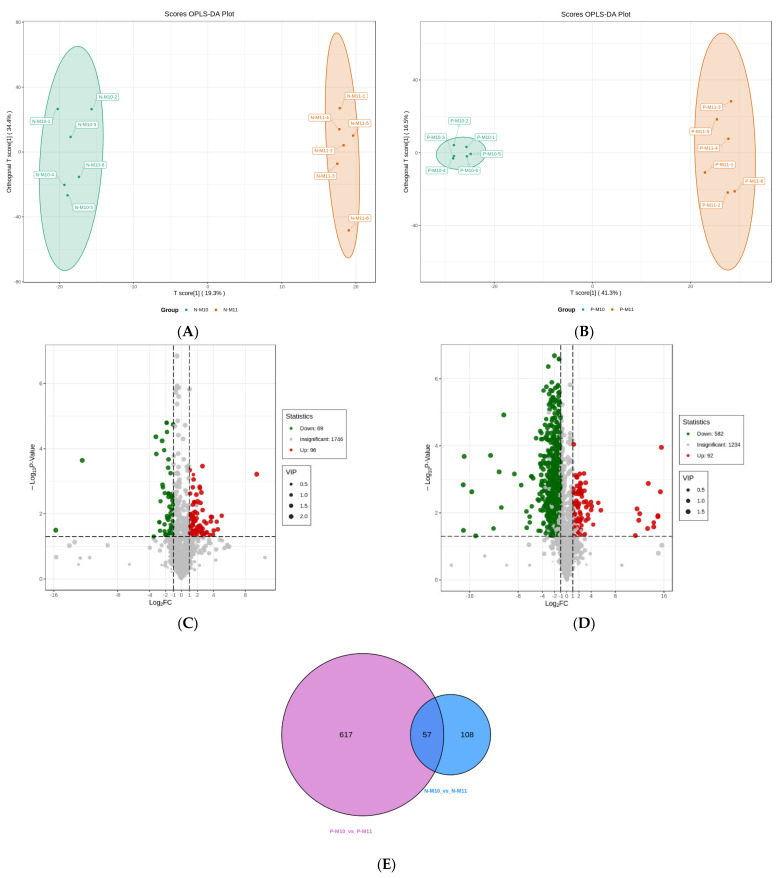
The score plots of OPLS-DA pairwise comparisons between N-M10 and N-M11 (**A**), P-M10 and P-M11 (**B**). Volcano plots of the differential metabolites between N-M10 and N-M11 (**C**), P-M10 and P-M11 (**D**). Venn diagram of differential metabolites (**E**). The T score of the score plot indicates the degree of variations between groups, and the orthogonal T score represents the degree of variations within the group. Percentages indicate how well a component explains the dataset. The volcano plot illustrates the distribution of the up-/downregulated differential metabolites. In the venn diagram, each comparison group is illustrated as a circle, and the number in it indicates the number of differential metabolites. The number in the overlapped area of the circles indicates the number of the common differential metabolites among all the comparison groups.

**Figure 4 foods-12-02178-f004:**
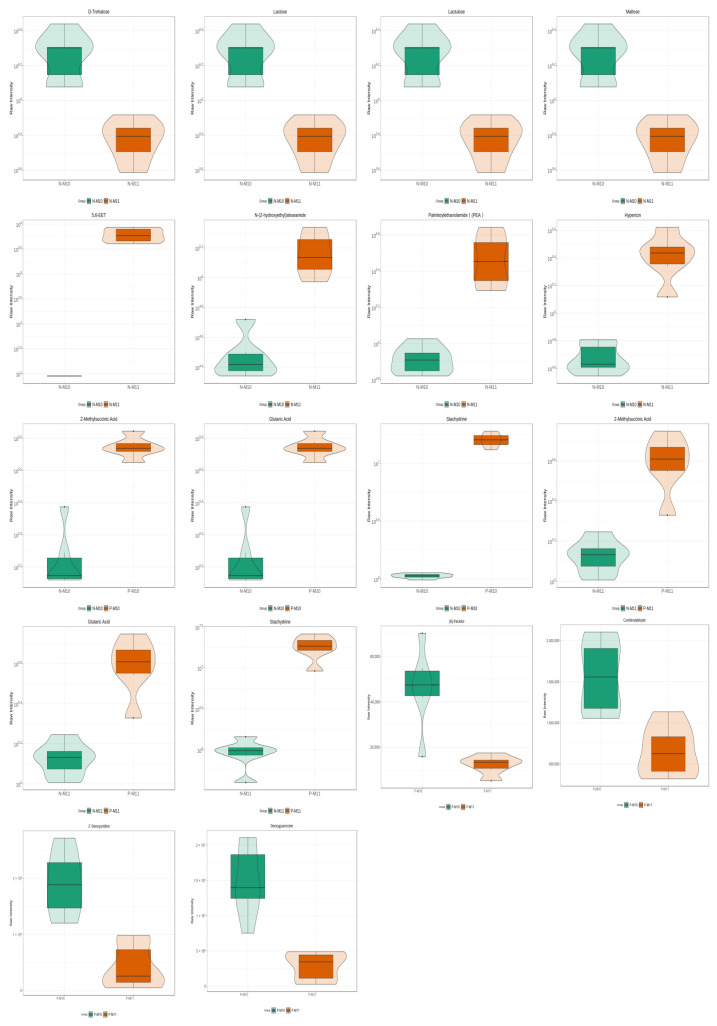
Representatives of the differential metabolites from *E. sinensis*. The box in the middle represents interquartile range. The 95% confidence interval is presented as a black line that penetrates through the box. The black line in the middle of the box is the median. The outer shape describes the density of distribution.

**Table 1 foods-12-02178-t001:** Free amino acid content of the meat and cream from *E. sinensis* (mg/g of wet weight).

	Meat				Cream			
	N-M10	P-M10	N-M11	P-M11	N-M10	P-M10	N-M11	P-M11
Asp ^1^	N.D.	0.02 ± 0.00 ^a^	0.01 ± 0.00 ^a^	0.01 ± 0.00 ^a^	1.08 ± 0.13 ^C^	1.41 ± 0.07 ^B^	1.72 ± 0.18 ^A^	1.50 ± 0.11 ^AB^
Thr	0.63 ± 0.07 ^a^	0.59 ± 0.06 ^ab^	N.D.	0.47 ± 0.05 ^b^	2.49 ± 0.22 ^A^	2.13 ± 0.16 ^A^	2.58 ± 0.32 ^A^	2.49 ± 0.33 ^A^
Ser	0.18 ± 0.02 ^b^	0.32 ± 0.07 ^a^	0.15 ± 0.04 ^b^	0.24 ± 0.03 ^ab^	1.34 ± 0.12 ^A^	1.41 ± 0.32 ^A^	1.25 ± 0.23 ^A^	0.98 ± 0.15 ^B^
Glu	0.17 ± 0.01 ^b^	0.32 ± 0.09 ^a^	0.12 ± 0.02 ^b^	0.31 ± 0.07 ^a^	2.95 ± 0.15 ^B^	2.91 ± 0.34 ^B^	3.80 ± 0.29 ^A^	3.88 ± 0.35 ^A^
Gly	1.69 ± 0.31 ^b^	3.59 ± 0.24 ^a^	1.50 ± 0.24 ^b^	3.80 ± 0.23 ^a^	1.32 ± 0.16 ^B^	1.09 ± 0.14 ^B^	1.67 ± 0.18 ^A^	1.89 ± 0.38 ^A^
Ala	2.20 ± 0.07 ^b^	4.87 ± 0.74 ^a^	1.78 ± 0.34 ^b^	5.07 ± 0.70 ^a^	2.81 ± 0.36 ^C^	2.72 ± 0.22 ^C^	4.86 ± 0.12 ^B^	5.50 ± 0.22 ^A^
Cys	0.04 ± 0.01 ^b^	0.08 ± 0.02 ^a^	0.02 ± 0.00 ^c^	0.07 ± 0.02 ^a^	0.26 ± 0.03 ^B^	0.32 ± 0.03 ^A^	0.25 ± 0.01 ^B^	0.28 ± 0.07 ^AB^
Val	0.14 ± 0.03 ^b^	0.27 ± 0.10 ^a^	0.11 ± 0.04 ^b^	0.29 ± 0.07 ^a^	1.68 ± 0.18 ^A^	1.92 ± 0.18 ^A^	1.99 ± 0.47 ^A^	2.04 ± 0.23 ^A^
Met	0.14 ± 0.01 ^bc^	0.31 ± 0.11 ^a^	0.10 ± 0.03 ^c^	0.23 ± 0.02 ^ab^	0.83 ± 0.03 ^A^	0.85 ± 0.12 ^A^	0.81 ± 0.06 ^A^	0.86 ± 0.04 ^A^
Ile	0.07 ± 0.02 ^bc^	0.15 ± 0.05 ^a^	0.05 ± 0.01 ^c^	0.13 ± 0.03 ^ab^	1.23 ± 0.19 ^A^	1.25 ± 0.35 ^A^	1.49 ± 0.09 ^A^	1.54 ± 0.22 ^A^
Leu	0.16 ± 0.01 ^ab^	0.29 ± 0.12 ^a^	0.12 ± 0.02 ^b^	0.22 ± 0.09 ^ab^	2.36 ± 0.29 ^A^	2.55 ± 0.44 ^A^	2.70 ± 0.28 ^A^	2.75 ± 0.12 ^A^
Tyr	0.07 ± 0.02 ^b^	0.27 ± 0.09 ^a^	0.09 ± 0.02 ^b^	0.14 ± 0.03 ^b^	1.36 ± 0.29 ^A^	1.32 ± 0.14 ^A^	1.22 ± 0.25 ^A^	1.16 ± 0.20 ^A^
Phe	0.10 ± 0.04 ^b^	0.29 ± 0.08 ^a^	0.11 ± 0.03 ^b^	0.23 ± 0.07 ^a^	1.42 ± 0.05 ^A^	1.57 ± 0.15 ^A^	1.42 ± 0.15 ^A^	1.45 ± 0.22 ^A^
Lys	0.11 ± 0.02 ^b^	0.46 ± 0.19 ^a^	0.13 ± 0.01 ^b^	0.51 ± 0.03 ^a^	2.12 ± 0.27 ^A^	2.34 ± 0.38 ^A^	2.74 ± 0.24 ^A^	2.09 ± 0.48 ^A^
His	0.14 ± 0.03 ^b^	0.31 ± 0.08 ^a^	0.12 ± 0.03 ^b^	0.25 ± 0.04 ^a^	0.76 ± 0.06 ^AB^	0.69 ± 0.17 ^B^	0.89 ± 0.09 ^A^	0.90 ± 0.07 ^A^
Arg	2.53 ± 0.19 ^b^	3.81 ± 0.18 ^a^	1.86 ± 0.33 ^c^	4.26 ± 0.17 ^a^	2.67 ± 0.27 ^A^	2.70 ± 0.29 ^A^	2.99 ± 0.14 ^A^	3.28 ± 0.52 ^A^
Pro	1.06 ± 0.06 ^b^	1.19 ± 0.22 ^b^	1.07 ± 0.05 ^b^	2.32 ± 0.17 ^a^	1.44 ± 0.11 ^B^	1.50 ± 0.24 ^B^	2.91 ± 0.40 ^A^	2.43 ± 0.32 ^A^
Essential AA	1.49 ± 0.09 ^c^	2.67 ± 0.37 ^a^	0.74 ± 0.13 ^d^	2.33 ± 0.13 ^b^	12.89 ± 0.45 ^C^	13.30 ± 0.37 ^BC^	14.62 ± 0.1 ^A^	14.12 ± 0.57 ^AB^
Total umami AA	0.17 ± 0.01 ^b^	0.34 ± 0.09 ^a^	0.13 ± 0.02 ^b^	0.32 ± 0.07 ^a^	4.03 ± 0.25 ^B^	4.32 ± 0.39 ^B^	5.52 ± 0.29 ^A^	5.38 ± 0.24 ^A^
Total sweet AA	5.76 ± 0.39 ^c^	10.56 ± 0.80 ^b^	4.50 ± 0.22 ^d^	11.90 ± 0.49 ^a^	9.40 ± 0.30 ^B^	8.85 ± 0.22 ^B^	13.27 ± 0.96 ^A^	13.29 ± 1.10 ^A^
Total bitter AA	3.39 ± 0.29 ^b^	5.89 ± 0.53 ^a^	2.60 ± 0.25 ^c^	6.12 ± 0.28 ^a^	13.07 ± 0.54 ^B^	13.87 ± 0.44 ^B^	15.03 ± 0.41 ^A^	14.91 ± 0.32 ^A^
Total AA	9.43 ± 0.31 ^c^	17.14 ± 0.55 ^b^	7.34 ± 0.04 ^d^	18.55 ± 0.64 ^a^	28.12 ± 0.15 ^B^	28.68 ± 0.99 ^B^	35.29 ± 0.67 ^A^	35.02 ± 1.20 ^A^

^1^ Letters in lower case mark the significant difference of each line of the meat samples (*p* < 0.05); letters in upper case mark the significant difference of each line of the cream samples (*p* < 0.05). Sweet AA includes Thr, Ser, Gly, Ala, and Pro. Umami AA includes Asp and Glu. Bitter AA includes Val, Met, Ile, Leu, Phe, Lys, His, and Arg. N.D. = not detected.

**Table 2 foods-12-02178-t002:** The fatty acid composition of the meat and cream from *E. sinensis* (% of total fatty acid).

	Meat				Cream			
	N-M10	P-M10	N-M11	P-M11	N-M10	P-M10	N-M11	P-M11
C_14:0_	0.35 ± 0.06	0.49 ± 0.03	0.38 ± 0.02	0.62 ± 0.03	0.94 ± 0.02	1.28 ± 0.05	0.77 ± 0.06	1.23 ± 0.03
C_15:0_	N.D.	0.32 ± 0.02	0.21 ± 0.01	0.28 ± 0.02	0.45 ± 0.03	0.73 ± 0.04	0.43 ± 0.02	0.56 ± 0.01
C_16:0_	14.50 ± 0.46	15.13 ± 0.11	14.15 ± 0.15	14.04 ± 0.40	20.17 ± 0.36	21.83 ± 0.42	19.26 ± 0.18	19.49 ± 0.12
C_17:0_	0.57 ± 0.08	0.66 ± 0.05	0.45 ± 0.02	0.54 ± 0.02	0.35 ± 0.09	0.48 ± 0.05	0.24 ± 0.01	0.35 ± 0.05
C_18:0_	7.56 ± 0.35	7.29 ± 0.40	6.71 ± 0.25	6.40 ± 0.01	2.49 ± 0.30	2.92 ± 0.24	2.49 ± 0.13	2.60 ± 0.06
C_20:0_	N.D.	N.D.	N.D.	N.D.	0.23 ± 0.02	N.D.	N.D.	0.21 ± 0.02
C_24:0_	12.47 ± 0.47	9.98 ± 0.14	12.48 ± 0.16	11.13 ± 0.13	1.71 ± 0.20	1.64 ± 0.10	1.42 ± 0.12	1.54 ± 0.16
SFA ^1^	35.45 ± 1.00 ^a^	33.87 ± 0.43 ^bc^	34.38 ± 0.55 ^b^	33.01 ± 0.21 ^c^	26.34 ± 0.26 ^B^	28.88 ± 0.72 ^A^	24.61 ± 0.42 ^C^	25.98 ± 0.44 ^B^
C_16:1 n7_	3.14 ± 0.17	3.20 ± 0.12	2.89 ± 0.02	2.79 ± 0.06	8.72 ± 0.15	7.81 ± 0.09	7.59 ± 0.10	7.71 ± 0.02
C_17:1 n7_	0.47 ± 0.06	0.51 ± 0.03	0.45 ± 0.06	0.44 ± 0.02	0.54 ± 0.05	0.50 ± 0.04	0.48 ± 0.02	0.52 ± 0.02
C_18:1 n9_	20.71 ± 0.51	22.74 ± 0.13	21.45 ± 0.11	21.75 ± 0.09	28.05 ± 0.17	27.45 ± 0.60	29.23 ± 0.16	29.06 ± 0.11
C_20:1 n9_	0.42 ± 0.08	0.51 ± 0.02	0.44 ± 0.01	1.58 ± 0.03	0.53 ± 0.05	0.74 ± 0.09	0.61 ± 0.02	0.92 ± 0.02
MUFA	24.74 ± 0.67 ^b^	26.96 ± 0.05 ^a^	25.24 ± 0.15 ^b^	26.56 ± 0.21 ^a^	37.84 ± 0.29 ^A^	36.50 ± 0.53 ^B^	37.92 ± 0.28 ^A^	38.22 ± 0.13 ^A^
C_18:2 n6_	17.27 ± 0.24	13.27 ± 0.22	18.11 ± 0.19	15.08 ± 0.09	25.43 ± 0.27	16.54 ± 0.43	26.47 ± 0.36	19.60 ± 0.19
C_18:3 n3_	1.66 ± 0.06	1.39 ± 0.06	1.34 ± 0.04	0.21 ± 0.02	2.39 ± 0.08	2.35 ± 0.19	1.70 ± 0.03	2.07 ± 0.01
C_20:2 n6_	1.31 ± 0.08	1.81 ± 0.11	1.37 ± 0.07	1.62 ± 0.10	0.90 ± 0.10	2.07 ± 0.29	1.09 ± 0.05	1.68 ± 0.10
C_20:3 n3_	0.35 ± 0.03	0.29 ± 0.33	0.34 ± 0.01	N.D.	0.31 ± 0.05	0.35 ± 0.48	0.32 ± 0.01	0.28 ± 0.03
C_20:4 n6_	4.62 ± 0.17	7.08 ± 0.23	4.56 ± 0.19	6.32 ± 0.17	1.08 ± 0.12	2.79 ± 0.20	1.11 ± 0.14	2.10 ± 0.09
C_22:6 n3_	6.30 ± 0.40	5.48 ± 0.26	6.45 ± 0.27	7.13 ± 0.20	0.93 ± 0.22	1.08 ± 0.06	0.89 ± 0.11	2.01 ± 0.19
PUFA	31.50 ± 0.64 ^a^	29.31 ± 0.07 ^c^	32.17 ± 0.37 ^a^	30.36 ± 0.35 ^b^	31.04 ± 0.22 ^A^	25.18 ± 0.61 ^C^	31.58 ± 0.63 ^A^	27.74 ± 0.26 ^B^
n3 PUFA	8.30 ± 0.46 ^a^	7.15 ± 0.27 ^b^	8.13 ± 0.27 ^a^	7.34 ± 0.19 ^b^	3.64 ± 0.22 ^B^	3.78 ± 0.14 ^B^	2.91 ± 0.11 ^C^	4.36 ± 0.16 ^A^
n6 PUFA	23.20 ± 0.23 ^b^	22.16 ± 0.21 ^c^	24.04 ± 0.11 ^a^	23.02 ± 0.17 ^b^	27.40 ± 0.28 ^B^	21.40 ± 0.57 ^D^	28.67 ± 0.54 ^A^	23.39 ± 0.33 ^C^
n3/n6	0.36 ± 0.02 ^a^	0.32 ± 0.02 ^b^	0.34 ± 0.01 ^ab^	0.32 ± 0.01 ^b^	0.13 ± 0.01 ^B^	0.18 ± 0.01 ^A^	0.10 ± 0.00 ^C^	0.18 ± 0.01 ^A^

^1^ Letters in lower case mark the significant difference of the SFA, MUFA, and PUFA of the meat samples (*p* < 0.05); letters in upper case mark the significant difference of the SFA, MUFA, and PUFA of the cream samples (*p* < 0.05). N.D. = not detected.

## Data Availability

The data presented in this study are available on request from the corresponding author.

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
