# Peer review of "Uncovering the Nutritive Profiles of Adult Male Chinese Mitten Crab (E. sinensis) Harvested from the Pond and Natural Water Area of Qin Lake Based on Metabolomics"

_foods, 2023, doi:10.3390/foods12112178_

Round 1

Reviewer 1 Report

Dear authors, here is my opinion about the article entitled “Uncovering the nutritive profiles of adult male Chinese mitten 2 crab (E. sinensis) harvested from the pond and natural water 3 area of Qin Lake based on metabolomics”, submitted for consideration for publication in Foods journal.

Concerning drafting quality, English should be improved, especially in the material and methods part. I recommend you revise English language by a native English speaker so that the paper become pleasant to read.

Regarding the scientific quality, I find the paper interesting and the metabolomic approach very relevant for this kind of studies.  This is why I encourage this kind of studies based on metabolomics, which is an efficient approach, which generates huge data on several samples with a huge saving of time.

Even if the study is interesting, I noted some imperfections in the design of the study, particularly regarding sample preparation, which is a very important step in metabolomic approaches, given the sensitivity of the metabolites (see details comments below). Authors should give more explanation about this point.

Also, at the first reading of the paper, I did not find interest in the comparison of crabs from two months of the same season (October vs November). I had to wait until the results and discussion part to understand that it was related to the maturity of the crabs, which is logical when you know it. That's why this explanation should be at the very beginning either in the introduction or in the material and method part.

In view of these elements, I find that the article deserves to be considered for publication in Foods journal, but after some minor corrections that I ask in my comments below.

1.     Introduction

Line 48: please give the meaning of PUFA as you did for MUFA and SFA. Even if it’s clear but some author may not understand what PUFA is.

Lines 53-69: you talk directly about using metabolomics to evaluate nutritional compounds in crabs. You should give a brief description of metabolomic approach (definition, its general use (different domain of biology), and why it is a useful approach to be considered by Scientifics), before talking about your specific use for the characterization of aquatic products’ nutritional quality

2.     Material and methods

2.1.         Sample preparation and sampling

Lines 77-79: Please reformulate the sentence.

Line 80: please replace the word “depended” by “dependent”. If you want to keep the word “depended” you should reformulate the sentence and say “the crabs from the natural water area of Qin Lake depended mostly on natural food”

Lines 81-84: Please reformulate the sentence “Twenty-one male crabs (200.0 81 ± 10.0 g) were collected from the pond in October, 2021 (P-M10), from natural water area 82 in October, 2021 (N-M10), from the pond in November, 2021 (P-M11) and from natural 83 water area in November, 2021 (N-M11), respectively.” . I had to read the sentence several times before I understand which crabs were from natural area and which ones were from artificial area, English should be improved  ? Also, how many crabs did you catch from the different areas and from the different months? Please clarify !!!

Lines 86-90: Do you mean that you analyzed 3 crabs + 18 crabs from each group ? that mean that you had not 21 crabs in total but 85 crabs. Please clarify the idea as I suggested in my previous comment.

Also you should explain if crabs were directly scarified after sampling from pound or natural areas. Were they transported to laboratory before dissection? How did they last alive between sampling and sacrifice ? You should also explain how samples were conserved, were they frozen,  or treated and analyzed directly after sacrifice ? You should know that the sample preparation and conservation is really important in metabolomic approach, because of the sensitivity of the addressed metabolite to any source of stress.

2.5.                 Wide targeted metabolomics analysis

Line 127: when you say “6 replicates in total “, do you mean you analyzed 6 replicates for each experimental group ? Reader will be confused until he arrived to the result part to understand what really was done. So please clarify this point in the material and method part

Lines 127-134: Were the samples lyophilized before proceeding to the metabolite extraction step? Generally, the lyophilization step is important because of the sensitivity of the metabolites.

Lines 147-154: the whole paragraph should be in the statistical analysis.

Line 154: You should explain how the fold change was calculated

3.     Results

Line 195: please replace “amino acid” by “amino acids”. It’s in the plural

Line 196: You mean “and their metabolites”?. Also correct “158 organic acid”  put it in the plural form

Line 197: please correct the plural form “117 nucleotide and its metabolomics” and replace “its metabolomics” by “their metabolites”

4.     Discussion

In the discussion part you should talk about the fact that samples were wot frozen nor dried after harvesting. This could impact the quality of the extracted metabolites. The results seem to be of good quality, but this point is important to be discussed in your paper.

Also, I think it will be interesting to discuss the environmental impact of pound culture as you discussed the interest of consumers for health and the choice of a healthy food.

5.     Conclusion

You should reformulate the conclusion part, so it will be organized as a brief summary part (brief introduction, and the most important results and a sentence to conclude the paper and finally a perspective sentence)

Concerning drafting quality, English should be improved, especially in the material and methods part. I recommend you revise English language by a native English speaker so that the paper become pleasant to read.

Reviewer 2 Report

Manuscript is novel enough, well written and presented by authors. However, there are some points that need to be addressed by authors. Concrete comments:

-Line 89-92:  In my opinion, the 18 edible parts of crabs are the real biological replicates and it is difficult to understand why the authors randomly mixed 3 edible parts to reduce the biological replicates from 18 to 6. The reduction in the number of samples does not only reduce informations but also leads to less robust statistical analyses. For all these reasons, it is important that the authors explain this point clearly.

- I suggest performing also this OPLS-DA model: PM10+PM11 vs N-M10+NM11 to analyse the global effect on metabolome of pond-reared versus naturally produced E. sinensis.
